# Vaccine effectiveness against SARS-CoV-2 infection or COVID-19 hospitalization with the Alpha, Delta, or Omicron SARS-CoV-2 variant: A nationwide Danish cohort study

Mie Agermose Gram[1]*, Hanne-Dorthe Emborg[1], Astrid Blicher Schelde[1], Nikolaj Ulrik Friis[1], Katrine Finderup Nielsen[1], Ida Rask Moustsen-Helms[1], Rebecca Legarth[2], Janni Uyen Hoa Lam[2], Manon Chaine[2], Aisha Zahoor Malik[2], Morten Rasmussen[3], Jannik Fonager[3], Raphael Niklaus Sieber[4], Marc Stegger[4], Steen Ethelberg[1,5], Palle Valentiner-Branth[1], Christian Holm Hansen[1,6]

1 Department of Infectious Disease Epidemiology and Prevention, Statens Serum Institut, Copenhagen, Denmark, 2 Division of Infectious Disease Preparedness, Data Integration and Analysis, Statens Serum Institut, Copenhagen, Denmark, 3 Department of Virus Research and Development Laboratory, Virus and Microbiological Special Diagnostics, Statens Serum Institut, Copenhagen, Denmark, 4 Department of Bacteria, Parasites, and Fungi, Statens Serum Institut, Copenhagen, Denmark, 5 Department of Public Health, Global Health Section, University of Copenhagen, Copenhagen, Denmark, 6 MRC International Statistics and Epidemiology Group, Department of Infectious Disease Epidemiology, London School of Hygiene & Tropical Medicine, London, United Kingdom

* MIAG@ssi.dk

**Data Availability Statement:** Data cannot be shared publicly due data protection regulation. Data

## Abstract

### Background

The continued occurrence of more contagious Severe Acute Respiratory Syndrome Coronavirus 2 (SARS-CoV-2) variants and waning immunity over time require ongoing reevaluation of the vaccine effectiveness (VE). This study aimed to estimate the effectiveness in 2 age groups (12 to 59 and 60 years or above) of 2 or 3 vaccine doses (BNT162b2 mRNA or mRNA-1273) by time since vaccination against SARS-CoV-2 infection and Coronavirus Disease 2019 (COVID-19) hospitalization in an Alpha-, Delta-, or Omicron-dominated period.

### Methods and findings

A Danish nationwide cohort study design was used to estimate VE against SARS-CoV-2 infection and COVID-19 hospitalization with the Alpha, Delta, or Omicron variant. Information was obtained from nationwide registries and linked using a unique personal identification number. The study included all previously uninfected residents in Denmark aged 12 years or above (18 years or above for the analysis of 3 doses) in the Alpha (February 20 to June 15, 2021), Delta (July 4 to November 20, 2021), and Omicron (December 21, 2021 to January 31, 2022) dominated periods. VE estimates including 95% confidence intervals (CIs) were calculated (1-hazard ratio·100) using Cox proportional hazard regression models with underlying calendar time and adjustments for age, sex, comorbidity, and geographical region. Vaccination status was included as a time-varying exposure. In the oldest age

are available from the Danish Health Data Authority for researchers who meet the criteria for access to confidential data. The data are available for research upon reasonable request and with permission from the Danish Data Protection Agency and the Danish Health Data Authority: https://sundhedsdatastyrelsen.dk/da/english/health_data_and_registers/research_services.

**Funding:** The author(s) received no specific funding for this work.

**Competing interests:** The authors have declared that no competing interests exist.

**Abbreviations:** CI, confidence interval; COVID-19, Coronavirus Disease 2019; CRS, Civil Registration System; Ct, cycle threshold; DVR, Danish Vaccination Registry; ICD-10, International Classification of Diseases, 10th revision; RT-PCR, reverse transcription polymerase chain reaction; SARS-CoV-2, Severe Acute Respiratory Syndrome Coronavirus 2; VE, vaccine effectiveness; VOC, variant of concern; WGS, whole-genome sequencing.

group, VE against infection after 2 doses was 90.7% (95% CI: 88.2; 92.7) for the Alpha variant, 82.3% (95% CI: 75.5; 87.2) for the Delta variant, and 39.9% (95% CI: 26.3; 50.9) for the Omicron variant 14 to 30 days since vaccination. The VE waned over time and was 73.2% (Alpha, 95% CI: 57.1; 83.3), 50.0% (Delta, 95% CI: 46.7; 53.0), and 4.4% (Omicron, 95% CI: −0.1; 8.7) >120 days since vaccination. Higher estimates were observed after the third dose with VE estimates against infection of 86.1% (Delta, 95% CI: 83.3; 88.4) and 57.7% (Omicron, 95% CI: 55.9; 59.5) 14 to 30 days since vaccination. Among both age groups, VE against COVID-19 hospitalization 14 to 30 days since vaccination with 2 or 3 doses was 98.1% or above for the Alpha and Delta variants. Among both age groups, VE against COVID-19 hospitalization 14 to 30 days since vaccination with 2 or 3 doses was 95.5% or above for the Omicron variant. The main limitation of this study is the nonrandomized study design including potential differences between the unvaccinated (reference group) and vaccinated individuals.

## Conclusions

Two vaccine doses provided high protection against SARS-CoV-2 infection and COVID-19 hospitalization with the Alpha and Delta variants with protection, notably against infection, waning over time. Two vaccine doses provided only limited and short-lived protection against SARS-CoV-2 infection with Omicron. However, the protection against COVID-19 hospitalization following Omicron SARS-CoV-2 infection was higher. The third vaccine dose substantially increased the level and duration of protection against infection with the Omicron variant and provided a high level of sustained protection against COVID-19 hospitalization among the +60-year-olds.

## Author summary

### Why was this study done?

- Previous studies have observed that the protection against infection and hospitalization from Coronavirus Disease 2019 (COVID-19) vaccination may differ between Severe Acute Respiratory Syndrome Coronavirus 2 (SARS-CoV-2) variants.

- Continued emergence of new variants and waning immunity by time since vaccination require ongoing evaluation of the vaccine effectiveness (VE) against infection and COVID-19 hospitalization to inform future vaccination strategies.

### What did the researchers do and find?

- We used register data on a nationwide cohort of Danish residents aged 12 years or above (18 years or above for the analysis of 3 doses) in 3 separate SARS-CoV-2 variant-dominated periods to estimate VE against infection and COVID-19 hospitalization with the Alpha, Delta, or Omicron variant.

- We observed high protection against SARS-CoV-2 infection with the Alpha (VE: 90.7%, 95% CI: 88.2; 92.7) and Delta (VE: 82.3%, 95% CI: 75.5; 87.2) variants after 2 vaccine doses and high levels of protection against hospitalization with all 3 variants.

- A third vaccine dose (VE: 57.7%, 95% CI: 55.9; 59.5) offered better protection against Omicron infection than 2 doses (VE: 39.9%, 95% CI: 26.3; 50.9) and boosted the levels of protection against COVID-19 hospitalization. There was less evidence of waning protection after 3 doses compared with 2 doses.

### What do these findings mean?

- Our findings indicate that a third dose is necessary to maintain protection against infection for longer time and to ensure a high level of protection against COVID-19 hospitalization with the Omicron variant.

- Further research with longer follow-up time is needed to evaluate the VE after more than 120 days to inform authorities about whether to administer further vaccine doses.

## Background

Mass vaccination of the population is a key strategy to manage the Coronavirus Disease 2019 (COVID-19) pandemic. However, breakthrough Severe Acute Respiratory Syndrome Coronavirus 2 (SARS-CoV-2) infections in vaccinated individuals still present a public health challenge [1,2]. Multiple studies have assessed COVID-19 vaccine effectiveness (VE) against SARS-CoV-2 infection and severe COVID-19 outcomes [3–9]. A previous systematic review and meta-regression demonstrated that the VE against SARS-CoV-2 infection and symptomatic disease decreased more than against severe disease 6 months after 2 doses [8]. However, all the included studies were carried out before the circulation of the Omicron variant [8]. The SARS-CoV-2 variants B.1.1.7 (Alpha), B.1.617.2 (Delta), and B.1.1.529 (Omicron) caused rapid increase of infections worldwide [10] and were classified as variants of concern (VOCs) by the World Health Organization [11]. Continued emergence of new variants and waning immunity by time since vaccination [2] require ongoing evaluation of the VE to inform future vaccination strategies. A Danish preprint study has estimated the protection of COVID-19 mRNA vaccines against infection or hospitalization with the Omicron variant and observed relatively poor protection against infection but high VE against COVID-19 hospitalization after the third dose [12]. However, previous studies have observed differences in the VE against SARS-CoV-2 infection with the Alpha, Delta, or Omicron variant [3–9], and only few large-scale studies have compared VE against all 3 variants [9]. Despite variation between the variants, a previous study that investigated the risk of hospitalization and death associated with the Delta or Omicron variant observed a significant variation with age [13]. The aim of this study was to estimate the effectiveness of 2 or 3 doses of the BNT162b2 mRNA (Pfizer/BioNTech) or mRNA-1273 (Moderna) vaccines against SARS-CoV-2 infection and COVID-19 hospitalization in an Alpha-, Delta-, or Omicron-dominant period by time since vaccination, and by 2 age groups, those aged 12 to 59 years and those 60 years and above.

## Methods

### Study design and setting

We conducted a nationwide cohort study in Denmark. All residents in Denmark are registered in the Danish Civil Registration System (CRS) with a unique personal identification number (CPR number), which is used in all national registries, enabling individual-level linkage between registries [14]. There were no missing data.

### The Danish COVID-19 vaccination program

The rollout of COVID-19 vaccines in Denmark was initiated on December 27, 2020. The BNT162b2 mRNA vaccine from Pfizer/BioNTech and the mRNA-1273 vaccine from Moderna are part of the Danish vaccination program. All residents aged 5 years or above are offered 2 vaccine doses, and those aged 18 years or above are offered a third vaccine dose 140 days (approximately 4.5 months) after the second dose. Very few people, particularly elderly nursing home residents and other vulnerable individuals, have also been offered a fourth dose. Denmark has continuously received vaccines during the COVID-19 vaccination rollout. However, relatively few vaccines were available in the initial phase. Hence, the Danish Health Authority determined the order in which population groups were offered vaccination. The populations initially prioritized for COVID-19 vaccination were the elderly, vulnerable citizens with increased risk of severe COVID-19, and frontline healthcare workers with the younger population invited during the summer of 2021 [15]. Broadly, those above the age of 60 years completed their primary vaccination schedule during spring 2021 when the Alpha variant was predominant, whereas the younger age groups were vaccinated during periods when subsequent variants were in circulation.

### SARS-CoV-2 testing

One of Denmark's main strategies for handling the COVID-19 epidemic was mass testing including unlimited access to free-of-charge SARS-CoV-2 reverse transcription polymerase chain reaction (RT-PCR) tests at either community testing facilities or hospitals, and rapid antigen tests at a community level. As part of the reopening of the country, a recent negative RT-PCR or rapid antigen test was required for unvaccinated individuals to access indoor public facilities (from March 1 to October 1, 2021, and again from November 11, 2021 to February 1, 2022). These initiatives have ensured a high testing rate for SARS-CoV-2, and the rate of RT-PCR testing in the Danish population has been among the highest in the world [10]. During the entire study period, it was recommended that a positive rapid antigen test was verified by RT-PCR.

### Identification of SARS-CoV-2 variants

RT-PCR tests for SARS-CoV-2 were analyzed at Statens Serum Institut or at the hospitals departments of clinical microbiology. Sequencing of the genome of SARS-CoV-2 was carried out by The Danish COVID-19 Genome Consortium, which was established in March 2020 with the purpose of assisting public health authorities by providing rapid genomic monitoring of the spread of SARS-CoV-2. Whole-genome sequencing (WGS) was performed by utilizing short read technology using the ARTIC v3 amplicon sequencing panel (https://artic.network) with spike-in of primers with slight modifications. Samples were sequenced on either the Next-Seq or NovaSeq platforms (Illumina), and consensus sequences were called using an in-house implementation of IVAR with a custom BCFtools command for consensus calling. Subvariants were called on all consensus sequences containing <3,000 ambiguous or missing sites using

Pangolin with the PangoLEARN assignment algorithm. Samples were randomly selected for WGS by an algorithm from all positive samples with cycle threshold (Ct) values below 35. N was called at unresolved positions with less than 10-fold coverage or if all bases were called at a position. Non-N ambiguous bases were called where there was less than 90% unambiguous variant calls. Sequences with 5 or more Non-N ambiguous bases were discarded as it could indicate contamination.

## Study population

The study population included all residents in Denmark aged 12 years or above (18 years or above for 3 doses) in an Alpha-, Delta-, and Omicron-dominant period. Individuals aged 5 to 11 years were offered vaccination later and vaccinated with a smaller dose than individuals aged 12 years or above (19). Therefore, we did not include individuals younger than 12 years in the study. Individuals vaccinated with other COVID-19 vaccines than BNT162b2 mRNA or mRNA-1273, and those with fewer than 19 days between the first and second dose, were censored at the time of vaccination. Those with fewer than 140 days between the second and third dose were censored at the time of the third dose. Individuals with an RT-PCR confirmed SARS-CoV-2 infection before the start of the study periods, and those without any tests during the 3 variant periods were excluded from the analyses. Only the first positive PCR-test was included.

## Assessment of exposure

All administered COVID-19 vaccines are registered in the Danish Vaccination Registry (DVR) on an individual level, identified by the CPR number [16]. Information on the date of vaccine administration and name of the vaccine product was retrieved from the DVR [16].

## Assessment of outcomes (SARS-CoV-2 infection and COVID-19 hospitalization)

The Danish Microbiology Database (MiBa) receives, in real time, copies of all laboratory test [17]. During the COVID-19 pandemic, community testing facilities were established across the nation, and in early 2021, it became mandatory for private vendors performing SARS-CoV-2 testing to report electronically to MiBa [17]. Data on all positive laboratory-confirmed RT-PCR tests were extracted from MiBa [17]. Information on rapid antigen tests was not included in this study due to moderate sensitivity in asymptomatic patients compared with RT-PCR [18].

All hospitalizations are registered in an individually identifiable format in the Danish National Patient Registry with date of admission and discharge as well as diagnoses coded according to the International Classification of Diseases, 10th revision (ICD-10) [19]. A COVID-19 hospitalization was defined as any new admission associated with ICD-10 primary diagnosis codes B342 or B972 [20] lasting at least 12 hours and occurring between 2 days before and 14 days after the sample date where SARS-CoV-2 infection with either the Alpha, Delta, or Omicron variant was detected.

## Covariates

Information on age, sex (male/female), comorbidity, and geographical region (Capital Region of Denmark/Central Denmark Region/Northern Denmark Region/Region Zealand/Region of Southern Denmark) was included as covariates for the association between COVID-19 vaccination and SARS-CoV-2 infection or COVID-19 hospitalization. Information on date of birth,

sex, and geographical region was extracted from the CRS registry [14]. Information on comorbidity (categorical variable with 4 levels indicating 0, 1, 2, or ≥3 comorbidities) was extracted from the Danish National Patient Registry [19]. The comorbidities were defined by ICD-10 codes. Each of the following comorbidities were recorded as either present or absent, and a count was calculated for each person indicating the total number of comorbidities: diabetes, adiposity, hematological and other cancers, neurological diseases, kidney diseases, cardiovascular diseases, chronic pulmonary diseases, respiratory diseases, and immune deficiency conditions.

## Statistical analysis

Characteristics of the study population were described using proportions and stratified by vaccination status. Time until SARS-CoV-2 infection (both asymptomatic and symptomatic) or COVID-19 hospitalization was analyzed in 3 periods where the relevant variant accounted for at least 75% of all WGS RT-PCR confirmed cases: Alpha (February 20 to June 15, 2021), Delta (July 4 to November 20, 2021), and Omicron (December 21, 2021 to January 31, 2022) (Fig 1).

Separate models were fitted to estimate the VE for the 2 outcomes, SARS-CoV-2 infection and COVID-19 hospitalization, and separately for the analysis of 2 or 3 doses. Event rates in

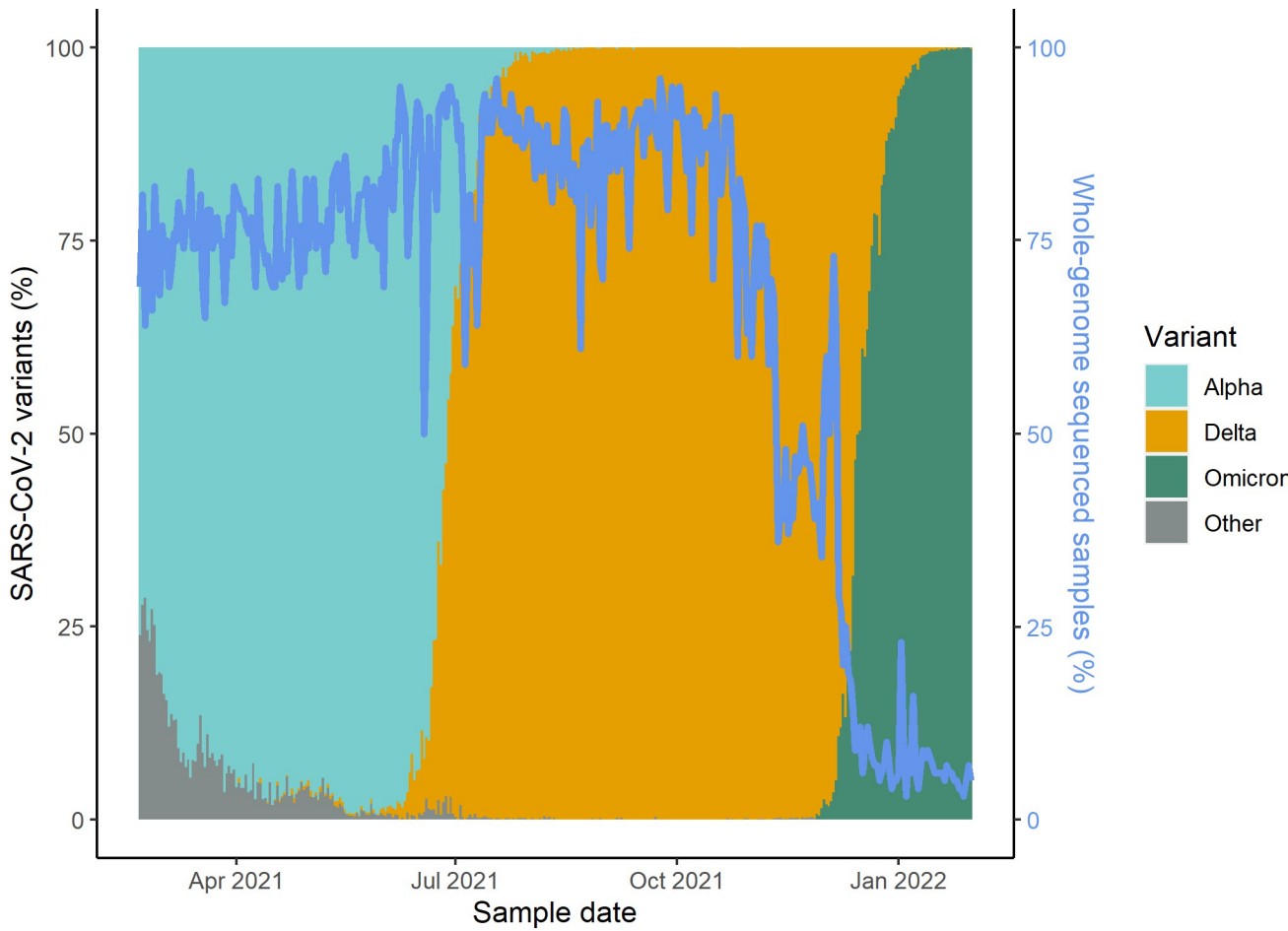

**Fig 1. Percentage of whole-genome sequenced samples and the distribution of SARS-CoV-2 variants by sample date.** The blue line represents the percentage of daily positive samples that were sequenced during the study period. The colored bars represent the distribution of the SARS-CoV-2 variants.

the vaccinated and unvaccinated exposure groups were compared using hazard ratios estimated in a Cox regression model adjusted for age (using a restricted cubic spline function with 5 knots), sex, comorbidity, and geographical region, with calendar time as the underlying time scale to control for temporal variations in the infection rate. VE was estimated as 1 minus the hazard ratio. Unadjusted VE estimates are available in the Supporting information (S3–S6 Tables).

Vaccinated individuals were followed from the start of the study or the date of assumed protection after the second or third vaccine dose, i.e., 14 days since vaccination. All individuals remained in follow-up until the date of SARS-CoV-2 infection, further vaccination (third dose among double vaccinated or fourth dose among triple vaccinated), death, emigration, or the end of study period, whichever occurred first. Unvaccinated individuals remained in follow-up from the start of the study and until the date of their first vaccination, death, emigration, SARS-CoV-2 infection, or end of the study, whichever occurred first.

Exposure status was categorized as either unvaccinated or vaccinated with the last dose administered in the past 14 to 30 days, 31 to 60 days, 61 to 90 days, 91 to 120 days, or >120 days. Time falling outside of these categories was not included in the analysis. Due to the timing of vaccination programs, VE estimates are not available for all combinations of doses, age groups, and variants. Data were analyzed using SAS version 9.4.

### Ethical considerations

According to Danish law, ethical approval is not required for anonymized aggregated register-based studies. The study adheres to the Strengthening the Reporting of Observational Studies in Epidemiology (STROBE) guidelines (S1 Table STROBE checklist) [21].

## Results

In the Delta-dominant period, fewer individuals aged 60 years or above were unvaccinated compared to the Alpha-dominant period. In the Omicron-dominant period, the majority in both age groups was vaccinated with 2 or 3 doses. In the Alpha- and Delta-dominant periods, the median age within each age group was slightly lower in the unvaccinated individuals compared to individuals vaccinated with 2 or 3 doses. Across all periods and age groups, the majority of vaccinated individuals were vaccinated with the BNT162b2 mRNA vaccine (87.4% to 98.4%) (Table 1).

During the Alpha and Delta periods, a high proportion of all SARS-CoV-2 infections was whole-genome sequenced with a median of 77% and 87% mid (<3,000 Ns) and high-quality (>150 Ns) genomes obtained using WGS (Fig 1) as previously described [22]. A lower median proportion was whole-genome sequenced during the Omicron period (6%) (Fig 1) due to the very high infection rate and a limited WGS capacity of 15,000 samples per week [23].

Older age groups were the first to be vaccinated with both the second and the third dose (Fig 2). Overall, the vaccination coverage was high in the population. The vaccination coverage on January 31, 2022 for 2 doses was 85% among individuals aged 12 to 59 and 95% in those 60 years or above. The vaccination coverage for 3 doses was 64% and 90% among individuals aged 18 to 59 and 60 years or above, respectively. Therefore, relatively few individuals contributed with follow-up time as unvaccinated (reference group) in the Delta- and especially Omicron-dominant periods (Fig 2).

### Vaccine effectiveness against SARS-CoV-2 infection after 2 mRNA doses

In the Alpha-dominant period, among individuals aged 60 or above, VE after 2 doses was 90.7% (95% CI: 88.2; 92.7) 14 to 30 days since vaccination. The estimate in the subsequent

**Table 1. Characteristics of the study population by vaccination status.**

| | Alpha February 20 –June 15, 2021 | | Delta July 4 –November 20, 2021 | | | | | | Omicron December 21, 2021 –January 31, 2022 | | | | | |
|---|---|---|---|---|---|---|---|---|---|---|---|---|---|---|
| | 60 years or above | | 12–59 years | | | 60 years or above | | | 12–59 years | | | 60 years or above | | |
| | Unvaccinated | Two doses | Unvaccinated | Two doses | Three doses* | Unvaccinated | Two doses | Three doses | Unvaccinated | Two doses | Three doses* | Unvaccinated | Two doses | Three doses |
| Number of individuals included | 652,324 | 409,103 | 961,946 | 1,725,954 | 62,373 | 22,097 | 545,811 | 81,470 | 179,417 | 1,189,665 | 967,356 | 10,899 | 47,998 | 468,051 |
| Mean number of PCR-tests | 2.4 | 3.2 | 1.1 | 3.2 | 4.2 | 3.0 | 2.3 | 2.6 | 2.2 | 2.3 | 2.8 | 2.5 | 2.1 | 2.2 |
| Sex, n (%) | | | | | | | | | | | | | | |
| Men | 310,372 (47.6) | 185,607 (45.4) | 466,587 (48.5) | 816,098 (47.3) | 15,857 (25.4) | 9,520 (43.1) | 256,517 (47.0) | 34,495 (42.3) | 88,718 (49.4) | 580,641 (48.8) | 453,703 (49.9) | 4,377 (40.2) | 23,399 (48.7) | 220,416 (47.1) |
| Women | 341,952 (52.4) | 223,496 (54.6) | 495,359 (51.5) | 909,856 (52.7) | 46,516 (74.6) | 12,577 (56.9) | 289,294 (53.0) | 46,975 (57.7) | 90,699 (50.6) | 609,024 (51.2) | 513,653 (53.1) | 6,522 (59.8) | 24,599 (51.3) | 247,635 (52.9) |
| Median age (IQR) | 68 (64; 74) | 73 (69; 77) | 29 (21;35) | 37 (24; 48) | 47 (38; 54) | (67 (62; 74) | 69 (64; 75) | 75 (67; 85) | 29 (21; 39) | 29 (19; 39) | 45 (32; 52) | 66 (62; 74) | 64 (61; 69) | 69 (64; 76) |
| Comorbidity | | | | | | | | | | | | | | |
| 0 | 434,393 (66.6) | 235,829 (57.7) | 856,474 (89.0) | 1,504,232 (82.7) | 42,930 (68.8) | 14,696 (66.5) | 349,519 (64.0) | 33,141 (40.7) | 158,707 (88.5) | 1,066,134 (89.6) | 824,671 (85.3) | 7,420 (68.1) | 32,887 (68.5) | 298,399 (63.8) |
| 1 | 150,111 (23.0) | 110,125 (26.9) | 94,493 (9.8) | 186,275 (10.8) | 13,429 (21.5) | 4,714 (21.3) | 130,174 (23.8) | 26,413 (32.4) | 18,266 (10.2) | 109,581 (9.2) | 116,757 (12.1) | 2,288 (21.0) | 9,827 (20.5) | 112,790 (24.1) |
| 2 | 50,259 (7.7) | 44,623 (10.9) | 9,568 (1.0) | 28,323 (1.6) | 4,042 (6.5) | 1,926 (8.7) | 47,803 (8.8) | 14,641 (18.0) | 2,065 (1.2) | 11,910 (1.0) | 20,214 (2.1) | 866 (7.9) | 3,784 (7.9) | 41,074 (8.8) |
| ≥3 | 17,561 (2.7) | 18,526 (4.5) | 1,411 (0.1) | 7,124 (0.4) | 1,972 (3.2) | 761 (3.4) | 18,315 (3.4) | 7,275 (8.9) | 379 (0.2) | 2,040 (0.2) | 5,714 (0.6) | 325 (3.0) | 1,500 (3.1) | 15,788 (3.4) |
| Geographical region, n (%) | | | | | | | | | | | | | | |
| Capital Region of Denmark | 176,864 (27.1) | 131,829 (32.2) | 311,087 (32.3) | 586,001 (34.0) | 21,406 (34.3) | 7,310 (33.1) | 161,616 (29.6) | 27,927 (34.3) | 60,514 (33.7) | 365,677 (30.7) | 307,584 (31.8) | 3,471 (31.8) | 10,988 (22.9) | 131,131 (28.0) |
| Central Denmark Region | 146,842 (22.5) | 87,326 (21.4) | 227,787 (23.7) | 390,339 (22.6) | 12,429 (19.9) | 3,972 (18.0) | 109,596 (20.1) | 15,948 (19.6) | 36,164 (20.2) | 280,723 (23.6) | 230,216 (23.8) | 2,075 (19.0) | 8,946 (18.6) | 104,139 (22.2) |
| Northern Denmark Region | 75,910 (11.6) | 41,863 (10.2) | 88,260 (9.2) | 164,472 (9.5) | 6,141 (9.8) | 2,046 (9.3) | 61,043 (11.2) | 7,803 (9.6) | 16,129 (9.0) | 124,314 (10.4) | 93,475 (9.7) | 1,013 (9.3) | 7,030 (14.6) | 49,428 (10.6) |
| Region Zealand | 98,613 (15.1) | 56,729 (13.9) | 117,568 (12.2) | 228,930 (13.3) | 8,426 (13.5) | 3,727 (16.9) | 90,114 (16.5) | 12,722 (15.6) | 28,042 (15.6) | 157,085 (13.2) | 130,626 (13.5) | 1,910 (17.5) | 9,459 (19.7) | 73,103 (15.6) |
| Region of Southern Denmark | 154,095 (23.6) | 91,356 (22.3) | 217,244 (22.6) | 356,212 (20.6) | 13,971 (22.4) | 5,042 (22.8) | 123,442 (22.6) | 17,070 (21.0) | 38,568 (21.5) | 261,866 (22.0) | 205,455 (21.2) | 2,430 (22.3) | 11,575 (24.1) | 110,250 (23.6) |
| Vaccine product, n (%) | | | | | | | | | | | | | | |
| BNT162b2 mRNA (Pfizer/BioNTech) | | 378,493 (92.5) | | 1,464,638 (84.9) | 61,346 (98.4) | | 499,496 (91.5) | 77,461 (95.1) | | 970,880 (81.6) | 845,026 (87.4) | | 42,823 (89.2) | 430,025 (91.9) |
| mRNA-1273 (Moderna) | | 30,610 (7.5) | | 261,316 (15.1) | 1,027 (1.6) | | 46,315 (8.5) | 4,009 (4.9) | | 218,785 (18.4) | 122,330 (12.6) | | 5,175 (10.8) | 38,026 (8.1) |

Individuals were able to contribute follow-up time in more than one time category.

*18–59 years for three doses.

time periods decreased with a VE of 73.2% (95% CI: 57.1; 83.3) >120 days since vaccination. In comparison, the 2-dose VE against SARS-CoV-2 infection with the Delta variant in individuals aged 60 years or above was 82.3% (95% CI: 75.5; 87.2) at 14 to 30 days since vaccination and decreased to 50.0% (95% CI: 46.7; 53.0) >120 days since vaccination. Slightly higher estimates were observed among individuals aged 12 to 59 years, where VE against SARS-CoV-2 infection with the Delta variant was 92.2% (95% CI: 91.8; 92.6) 14 to 30 days since vaccination and decreased to 64.8% (95% CI: 63.9; 65.8) >120 days since vaccination. Markedly lower 2-dose VE estimates were observed against SARS-CoV-2 infection with the Omicron variant for both age groups. Among individuals aged 12 to 59 years, VE was estimated at 40.0% (95%

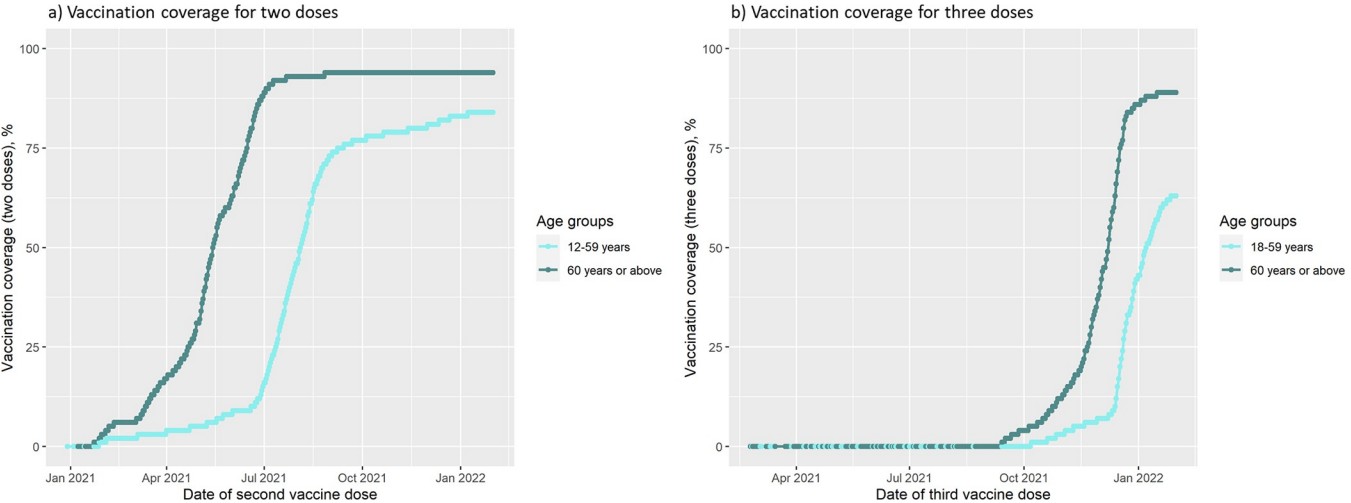

**Fig 2. Percentage vaccinated with 2 doses or 3 doses of BNT162b2 mRNA or mRNA-1273 by age groups.** Panel a represents the vaccination coverage during the study period for 2 doses. Panel b represents the vaccination coverage during the study period for 3 doses.

CI: 38.6; 41.3) 14 to 30 days after vaccination and decreased to 12.6% (95% CI: 12.0; 13.39) >120 days since vaccination. Similar estimates but with wider CIs were observed among individuals aged 60 years or above (Table 2 and Fig 3).

## Vaccine effectiveness against SARS-CoV-2 infection after 3 mRNA doses

Similar to the VE estimates after 2 doses, the VE after 3 doses was markedly lower against SARS-CoV-2 infection with the Omicron variant than the Delta variant. Nonetheless, the VE against the Omicron variant was higher after 3 than after 2 doses and with less waning by time since vaccination (Tables 2 and 3). Among individuals aged 18 to 59 years, only a relatively small number received a third dose during the Delta period, including those particularly exposed, e.g., in the healthcare professions, or those at high risk of progression to serious disease. In this group, the VE against SARS-CoV-2 infection with the Delta variant was 89.5% (95% CI: 87.6; 91.1) 14 to 30 days since vaccination and 85.1% (95% CI: 66.8; 93.3) 61 to 90 days since vaccination. Similar estimates were observed among individuals aged 60 years or above. In comparison, for individuals aged 18 to 59 years, VE after 3 doses against SARS-CoV-2 infection with the Omicron variant was 55.1% (95% CI: 54.6; 55.5) 14 to 30 days since vaccination and 52.3% (95% CI: 48.0; 56.2) >120 days since vaccination. Among individuals aged 60 years or above, VE against SARS-CoV-2 infection with the Omicron variant after 3 doses was 57.7% (95% CI: 55.9; 59.5) 14 to 30 days since vaccination and 53.2% (95% CI: 49.6; 56.6) >120 days since vaccination (Table 3 and Fig 3).

## Vaccine effectiveness against COVID-19 hospitalization after 2 mRNA doses

A high VE against COVID-19 hospitalization after 2 doses was observed for both the Alpha and Delta variants. However, lower protection was observed against COVID-19 hospitalization following infection with the Omicron variant (Table 4). Among individuals aged 60 years or above, VE against COVID-19 hospitalization with the Alpha and Delta variants after 2 doses was 98.1% (95% CI: 94.7; 99.3) and 100% (95% CI was not estimated as no hospital admissions were observed), respectively, 14 to 30 days since vaccination. The estimates >120

**Table 2. Adjusted VE of 2 doses BNT162b2 mRNA or mRNA-1273 against SARS-CoV-2 infection with the Alpha, Delta, or Omicron variant by age groups (12–59 years and 60 years or above).**

| | Alpha | | | | | Delta | | | | | Omicron | | | | |
|---|---|---|---|---|---|---|---|---|---|---|---|---|---|---|---|
| | Population | Person-years | Cases | Adjusted VE | 95% CI | Population | Person-years | Cases | Adjusted VE | 95% CI | Population | Person-years | Cases | Adjusted VE | 95% CI |
| **12–59 years** | | | | | | | | | | | | | | | |
| Unvaccinated | | | | | | 961,947 | 143,400 | 43,581 | 1 (reference) | | 179,417 | 15,470 | 96,160 | 1 (reference) | |
| **Time since vaccination** | | | | | | | | | | | | | | | |
| 14–30 days | | | | | | 1,600,382 | 74,050 | 1,624 | 92.2 | 91.8; 92.6 | 61,480 | 2,147 | 8,252 | 40.0 | 38.6; 41.3 |
| 31–60 days | | | | | | 1,598,449 | 129,640 | 3,718 | 88.1 | 87.7; 88.5 | 63,919 | 3,019 | 16,502 | 31.9 | 30.7; 33.0 |
| 61–90 days | | | | | | 1,581,085 | 123,574 | 7,873 | 80.8 | 80.2; 81.2 | 57,597 | 2,158 | 9,759 | 32.3 | 30.9; 33.7 |
| 91–120 days | | | | | | 1,400,902 | 85,887 | 14,950 | 72.2 | 71.5; 72.8 | 221,164 | 7,017 | 22,911 | 31.3 | 30.3; 32.4 |
| >120 days | | | | | | 750,393 | 59,184 | 10,360 | 64.8 | 63.9; 65.8 | 1,076,044 | 56,890 | 276,075 | 12.6 | 12.0; 13.3 |
| **60 years or above** | | | | | | | | | | | | | | | |
| Unvaccinated | 652,324 | 111,191 | 4,462 | 1 (reference) | | 22,097 | 6,895 | 1,113 | 1 (reference) | | 10,899 | 1,051 | 3,351 | 1 (reference) | |
| **Time since vaccination** | | | | | | | | | | | | | | | |
| 14–30 days | 407,513 | 16,797 | 78 | 90.7 | 88.2; 92.7 | 199,220 | 6,996 | 59 | 82.3 | 75.5; 87.2 | 1,341 | 48 | 96 | 39.9 | 26.3; 50.9 |
| 31–60 days | 323,594 | 16,190 | 144 | 83.2 | 79.7; 86.1 | 360,044 | 21,964 | 393 | 74.4 | 70.1; 78.2 | 1,290 | 61 | 136 | 39.0 | 27.6; 48.7 |
| 61–90 days | 116,308 | 7,274 | 119 | 73.0 | 66.8; 78.1 | 447,290 | 34,207 | 687 | 77.3 | 74.4; 79.9 | 1,072 | 43 | 103 | 25.2 | 9.0; 38.6 |
| 91–120 days | 58,348 | 3,467 | 49 | 82.6 | 76.5; 87.2 | 496,192 | 38,465 | 1,159 | 69.6 | 66.5; 72.4 | 1,750 | 77 | 173 | 24.0 | 11.4; 34.8 |
| >120 days | 35,699 | 1,445 | 23 | 73.2 | 57.1; 83.3 | 534,325 | 92,247 | 12,198 | 50.0 | 46.7; 53.0 | 45,835 | 1,781 | 4,722 | 4.4 | −0.1; 8.7 |

CI, confidence interval; VE, vaccine effectiveness.

Person-years in days. VE estimates adjusted for underlying calendar time, age, sex, comorbidity (categorical variable with 4 levels indicating 0, 1, 2, or $\geq 3$ comorbidities), and geographical region. Individuals were able to contribute follow-up time in more than one time category and (if vaccinated during the study period) to both the analysis of VE after 2 or 3 doses.

days since vaccination were 96.5% (Alpha, 95% CI: 73.4; 99.5) and 87.5% (Delta, 95% CI: 85.6; 89.2). For the Delta variant, similar estimates were observed among individuals aged 12 to 59 years (Table 4). Among individuals aged 12 to 59 years, VE against COVID-19 hospitalization following infection with the Omicron variant was 96.2% (95% CI: 72.9; 99.5) 14 to 30 days since vaccination and 77.6% (95% CI: 72.6; 81.6) >120 days since vaccination. It was not possible to estimate 2-dose VE against COVID-19 hospitalization following infection with the Omicron variant among individuals aged 60 years or above due to few cases, and since the majority of this group had already received a third vaccine dose at this time (Table 4 and Fig 4).

## Vaccine effectiveness against COVID-19 hospitalization after 3 mRNA doses

For both age groups, small absolute differences were observed in the VE estimates of 3 doses against COVID-19 hospitalization between the Delta or Omicron variant. However, it is difficult to compare the estimates between the Delta and Omicron variant because the available time interval after vaccination during the Delta period was shorter. Among the relatively small

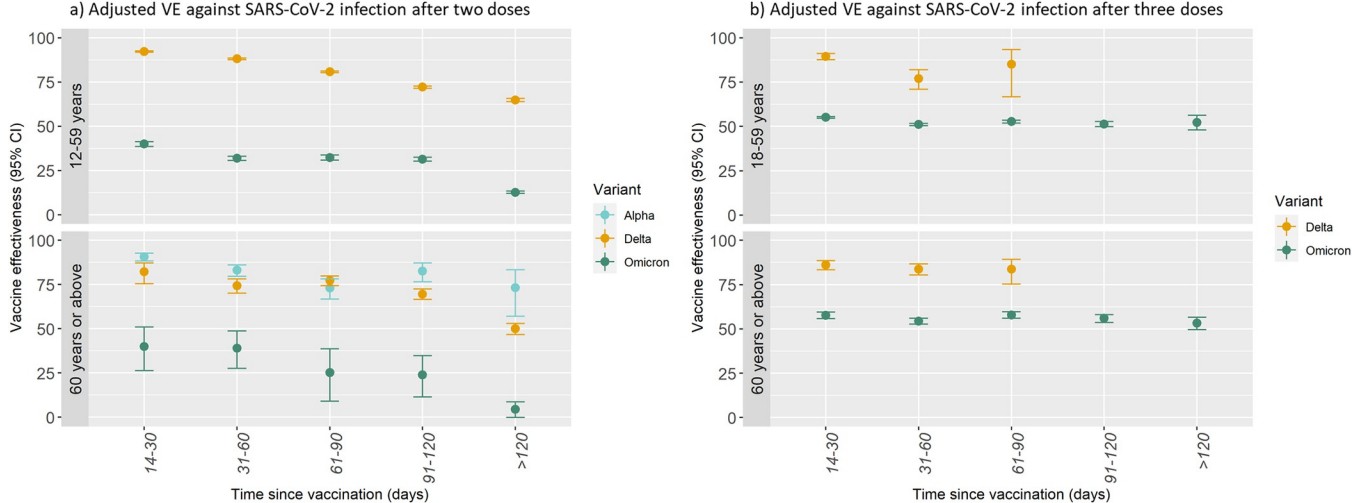

**Fig 3. Adjusted VE against SARS-CoV-2 infection after 2 or 3 doses of BNT162b2 mRNA or mRNA-1273 by SARS-CoV-2 variant and age group.** Panel a represents VE against SARS-CoV-2 infection after 2 doses. Panel b represents VE against SARS-CoV-2 infection after 3 doses. The VE estimates are adjusted for underlying calendar time, age, sex, comorbidity, and geographical region. CI, confidence interval; SARS-CoV-2, Severe Acute Respiratory Syndrome Coronavirus 2; VE, vaccine effectiveness.

group of individuals aged 18 to 59 years who were vaccinated with a third dose in the autumn of 2021, VE against COVID-19 hospitalization following infection with the Delta variant was 93.2% (95% CI: 81.7; 97.5) 14 to 30 days and 61.5% (95% CI: 23.4; 80.6) 31 to 60 days since

**Table 3. Adjusted VE of 3 doses BNT162b2 mRNA or mRNA-1273 against SARS-CoV-2 infection with the Delta or Omicron variant by age groups (18–59 years and 60 years or above).**

| | Delta | | | | | Omicron | | | | |
|---|---|---|---|---|---|---|---|---|---|---|
| | Population | Person-years | Cases | Adjusted VE | 95% CI | Population | Person-years | Cases | Adjusted VE | 95% CI |
| 18–59 years | | | | | | | | | | |
| Unvaccinated | 757,872 | 110,504 | 34,412 | 1 (reference) | | 144,946 | 12,652 | 74,627 | 1 (reference) | |
| Time since vaccination | | | | | | | | | | |
| 14–30 days | 62,373 | 2,040 | 149 | 89.5 | 87.6; 91.1 | 880,288 | 34,122 | 103,590 | 55.1 | 54.6; 55.5 |
| 31–60 days | 19,416 | 502 | 69 | 77.0 | 70.9; 81.9 | 631,525 | 20,080 | 87,334 | 51.1 | 50.4; 51.7 |
| 61–90 days | 2,736 | 47 | 6 | 85.1 | 66.8; 93.3 | 91,301 | 5,505 | 15,540 | 52.7 | 51.8; 53.6 |
| 91–120 days | | | | | | 44,271 | 1,329 | 5,322 | 51.3 | 49.8; 52.7 |
| >120 days | | | | | | 4,259 | 144 | 580 | 52.3 | 48.0; 56.2 |
| 60 years or above | | | | | | | | | | |
| Unvaccinated | 22,097 | 6,895 | 1,113 | 1 (reference) | | 10,899 | 1,051 | 3,351 | 1 (reference) | |
| Time since vaccination | | | | | | | | | | |
| 14–30 days | 81,470 | 3,031 | 156 | 86.1 | 83.3; 88.4 | 335,215 | 12,702 | 13,477 | 57.7 | 55.9; 59.5 |
| 31–60 days | 45,216 | 2,376 | 162 | 83.8 | 80.4; 86.7 | 390,527 | 21,077 | 34,137 | 54.4 | 52.7; 56.0 |
| 61–90 days | 14,015 | 225 | 26 | 83.7 | 75.2; 89.2 | 176,457 | 6,999 | 10,366 | 57.9 | 56.1; 59.6 |
| 91–120 days | | | | | | 75,122 | 3,723 | 5,088 | 56.0 | 53.7; 58.1 |
| >120 days | | | | | | 31,522 | 1,101 | 2,130 | 53.2 | 49.6; 56.6 |

CI, confidence interval; VE, vaccine effectiveness.

VE estimates adjusted for underlying calendar time, age, sex, comorbidity (categorical variable with four levels indicating 0, 1, 2, or ≥3 comorbidities), and geographical region. Individuals were able to contribute follow-up time in more than one time category and (if vaccinated during the study period) to both the analysis of VE after 2 or 3 doses.

**Table 4. Adjusted VE of 2 doses BNT162b2 mRNA or mRNA-1273 against COVID-19 hospitalization following infection with the Alpha, Delta, or Omicron variant by age groups (12–59 years and 60 years or above).**

| | Alpha | | | | | Delta | | | | | Omicron | | | | |
|---|---|---|---|---|---|---|---|---|---|---|---|---|---|---|---|
| | Population | Person-years | Cases | Adjusted VE | 95% CI | Population | Person-years | Cases | Adjusted VE | 95% CI | Population | Person-years | Cases | Adjusted VE | 95% CI |
| **12–59 years** | | | | | | | | | | | | | | | |
| Unvaccinated | | | | | | 961,947 | 143,400 | 724 | 1 (reference) | | 179,417 | 15,470 | 263 | 1 (reference) | |
| Time since vaccination | | | | | | | | | | | | | | | |
| 14–30 days | | | | | | 1,600,382 | 74,050 | 3 | 99.5 | 98.4; 99.8 | 61,480 | 2,147 | 1 | 96.2 | 72.9; 99.5 |
| 31–60 days | | | | | | 1,598,449 | 129,640 | 7 | 99.4 | 98.7; 99.7 | 63,919 | 3,019 | 8 | 79.0 | 57.5; 89.6 |
| 61–90 days | | | | | | 1,581,085 | 123,574 | 7 | 99.2 | 98.4; 99.6 | 57,597 | 2,158 | 8 | 70.5 | 40.3; 85.5 |
| 91–120 days | | | | | | 1,400,902 | 85,887 | 25 | 97.9 | 96.9; 98.6 | 221,164 | 7,017 | 11 | 85.8 | 73.8; 92.3 |
| >120 days | | | | | | 750,393 | 59,184 | 77 | 93.5 | 91.6; 95.0 | 1,076,044 | 56,890 | 168 | 77.6 | 72.6; 81.6 |
| **60 years or above** | | | | | | | | | | | | | | | |
| Unvaccinated | 652,324 | 111,191 | 615 | 1 (reference) | | 22,097 | 6,895 | 276 | 1 (reference) | | | | | | |
| Time since vaccination | | | | | | | | | | | | | | | |
| 14–30 days | 407,513 | 16,797 | 4 | 98.1 | 94.7; 99.3 | 199,220 | 6,996 | 0 | 100.0 | * | | | | | |
| 31–60 days | 323,594 | 16,190 | 15 | 94.0 | 89.6; 96.6 | 360,044 | 21,964 | 8 | 97.5 | 94.8; 98.8 | | | | | |
| 61–90 days | 116,308 | 7,274 | 19 | 86.5 | 77.4; 91.9 | 447,290 | 34,207 | 18 | 97.7 | 96.3; 98.6 | | | | | |
| 91–120 days | 58,348 | 3,467 | 5 | 94.0 | 84.9; 97.6 | 496,192 | 38,465 | 32 | 97.2 | 95.9; 98.1 | | | | | |
| >120 days | 35,699 | 1,445 | 1 | 96.5 | 73.4; 99.5 | 534,325 | 92,247 | 738 | 87.5 | 85.6; 89.2 | | | | | |

CI, confidence intervals; VE, vaccine effectiveness.

VE estimates adjusted for underlying calendar time, age, sex, comorbidity (categorical variable with 4 levels indicating 0, 1, 2, or ≥3 comorbidities), and geographical region. Individuals were able to contribute follow-up time in more than one time category and (if vaccinated during the study period) to both the analysis of VE after 2 or 3 doses.

*It was not possible within the model to estimate a 95% CI for the estimated VE against COVID-19 hospitalization with the Delta variant 14–30 days after the second dose as no COVID-19-related hospitalization were observed.

vaccination. In individuals aged 60 years or above, VE against COVID-19 hospitalization following infection with the Delta variant was 97.2% (95% CI: 94.6; 98.5) 14 to 30 days and 91.7% (95% CI: 78.7; 96.7) 61 to 90 days since vaccination. In comparison, in individuals aged 18 to 59 years, VE against COVID-19 hospitalization following infection with the Omicron variant was 95.5% (95% CI: 93.5; 96.8) 14 to 30 days since vaccination. From then on, a gradual decline in VE was observed reaching 67.5% (95% CI: 50.4; 78.7) 91 to 120 days since vaccination. A smaller decrease in the VE against COVID-19 hospitalization following infection with the Omicron variant was observed among individuals aged 60 years or above with an estimated VE of 96.7% (95% CI: 95.6; 97.6) 14 to 30 days and 83.3% (95% CI: 77.3; 87.8) >120 days since vaccination (Table 5 and Fig 4).

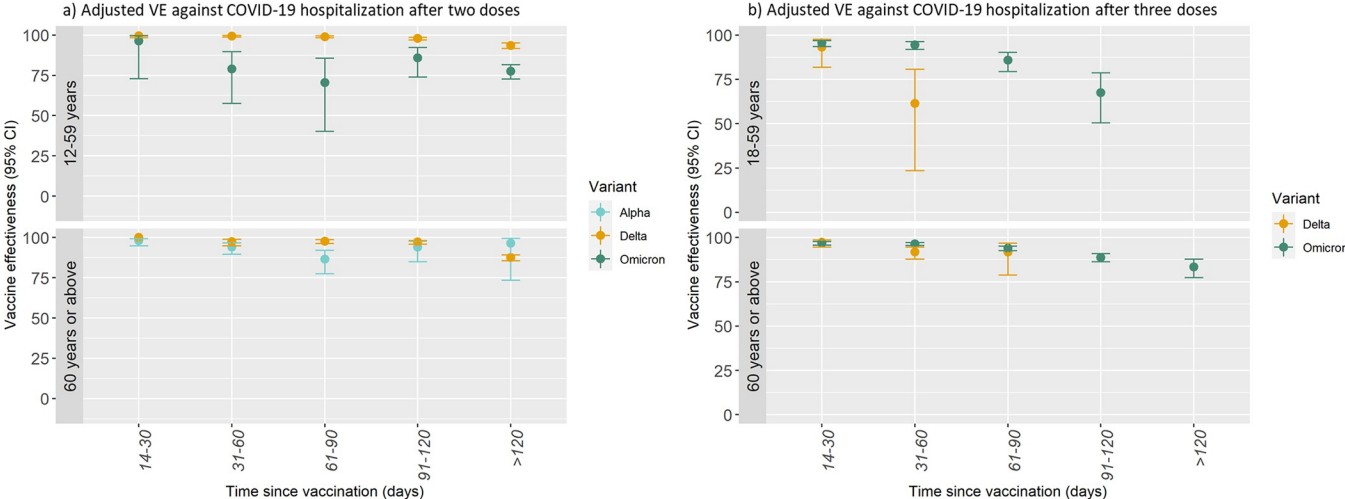

**Fig 4. Adjusted VE against COVID-19-related hospitalization after 2 or 3 doses of BNT162b2 mRNA or mRNA-1273 by SARS-CoV-2 variant and age group.** Panel a represents VE against COVID-19 hospitalization after 2 doses. Panel b represents adjusted VE against COVID-19 hospitalization after 3 doses. The VE estimates are adjusted for underlying calendar time, age, sex, comorbidity, and geographical region. CI, confidence interval; COVID-19, Coronavirus Disease 2019; SARS-CoV-2, Severe Acute Respiratory Syndrome Coronavirus 2; VE, vaccine effectiveness.

## Discussion

Compared to unvaccinated individuals, vaccination with 2 or 3 doses of BNT162b2 mRNA or mRNA-1273 was associated with protection against infection with the Alpha, Delta, or Omicron SARS-CoV-2 variant, peaking 14 to 30 days since vaccination with either 2 or 3 doses. However, the protection against infection with the Omicron variant was markedly lower compared to the protection against infection with the Alpha or Delta variants. The protection against infection afforded by vaccination after 2 doses decreased with time since vaccination. However, the VE against infection decreased less over time after 3 doses. A high level of protection and less pronounced waning were observed against COVID-19 hospitalization following infection with the Alpha and Delta variants after 2 doses. However, the third dose contributed to greater levels of protection against COVID-19 hospitalization following infection especially with the Omicron variant. It was unexpected that individuals aged 12 to 59 years had slightly lower VE against infection with the Omicron variant in all time intervals compared to individuals aged 60 years or above. This may be explained by biases introduced in nonrandomized studies including differences in behavior between unvaccinated and vaccinated individuals [24].

An overview of results from previous studies within this field of research is shown in tabular form in the Supporting information (S2 Table). Our results regarding the VE against SARS-CoV-2 infection with the Alpha and Delta variants align well with those of a test-negative case–control study from England that observed a slightly higher VE against SARS-CoV-2 infection with the Alpha variant compared to the Delta variant after 2 doses of the BNT162b2 mRNA vaccine [3]. They observed a VE against SARS-CoV-2 infection with the Alpha and Delta variants of 93.7% (95% CI: 91.6; 95.3) and 88.0% (95% CI: 85.3; 90.1), respectively [3]. However, in another study from the United Kingdom, similar VE estimates against infection with the Alpha (78% (95% CI: 68; 84)) or Delta (80% (95% CI: 77; 83)) variants after 2 doses of BNT162b2 mRNA were observed [4]. Previous studies have observed markedly higher protection against infection with the Delta variant compared to the Omicron variant [5–7]. A test-negative case–control study from Southern California found that VE against infection with the

**Table 5. Adjusted VE of 3 doses BNT162b2 mRNA or mRNA-1273 against COVID-19 hospitalization following infection with the Delta or Omicron variant by age groups (18–59 years and ≥60 years).**

| | Delta | | | | | Omicron | | | | |
|---|---|---|---|---|---|---|---|---|---|---|
| | Population | Person-years | Cases | Adjusted VE | 95% CI | Population | Person-years | Cases | Adjusted VE | 95% CI |
| **18–59 years** | | | | | | | | | | |
| Unvaccinated | 757,872 | 110,504 | 717 | 1 (reference) | | 144,946 | 12,652 | 255 | 1 (reference) | |
| Time since vaccination | | | | | | | | | | |
| 14–30 days | 62,373 | 2,040 | 4 | 93.2 | 81.7; 97.5 | 880,288 | 34,122 | 39 | 95.5 | 93.5; 96.8 |
| 31–60 days | 19,416 | 502 | 9 | 61.5 | 23.4; 80.6 | 631,525 | 20,080 | 41 | 94.5 | 91.9; 96.2 |
| 61–90 days | | | | | | 91,301 | 5,505 | 37 | 85.8 | 79.4; 90.2 |
| 91–120 days | | | | | | 44,271 | 1,329 | 32 | 67.5 | 50.4; 78.7 |
| >120 days | | | | | | | | | | |
| **60 years or above** | | | | | | | | | | |
| Unvaccinated | 22,097 | 6,895 | 276 | 1 (reference) | | 10,899 | 1,051 | 236 | Reference | |
| Time since vaccination | | | | | | | | | | |
| 14–30 days | 81,470 | 3,031 | 10 | 97.2 | 94.6; 98.5 | 335,215 | 12,702 | 65 | 96.7 | 95.6; 97.6 |
| 31–60 days | 45,216 | 2,376 | 32 | 91.8 | 87.7; 94.6 | 390,527 | 21,077 | 163 | 96.4 | 95.5; 97.1 |
| 61–90 days | 14,015 | 225 | 5 | 91.7 | 78.7; 96.7 | 176,457 | 6,999 | 159 | 94.0 | 92.5; 95.1 |
| 91–120 days | | | | | | 75,122 | 3,723 | 213 | 88.7 | 86.2; 90.8 |
| >120 days | | | | | | 31,522 | 1,101 | 95 | 83.3 | 77.3; 87.8 |

CI, confidence interval; VE, vaccine effectiveness.

VE estimates adjusted for underlying calendar time, age, sex, comorbidity (categorical variable with four levels indicating 0, 1, 2, or ≥3 comorbidities), and geographical region. Individuals were able to contribute follow-up time in more than one time category and (if vaccinated during the study period) to both the analysis of VE after 2 or 3 doses.

Delta variant after 2 doses of mRNA-1273 was high and waned slowly with VE of 80.2% (95% CI: 68.2; 87.7) and 61.3% (95% CI: 55.0; 66.7) 14 to 90 days and >270 days since vaccination, respectively [7]. Furthermore, they observed VE against hospitalization with the Delta variant of ≥99% after 2 or 3 doses (8). Similar to our study, a test-negative case–control study from England also observed waning over time and that a third dose contributed to greater levels of protection [6]. They observed that VE against infection with the Delta variant after 2 doses BNT162b2 mRNA was 90.9% (95% CI: 89.6; 92.0) 2 to 4 weeks since vaccination and declined to 62.7% (95% CI: 61.6; 63.7) ≥25 weeks since vaccination. After the third dose, the VE against infection with the Delta variant was 95.1% (95% CI: 94.8; 95.4) 2 to 4 weeks since vaccination and declined to 89.9% (95% CI: 89.2; 90.5) ≥10 weeks since vaccination [6]. The VE was higher with the mRNA-1273 vaccine [6].

For the Omicron variant, the study from Southern California observed only a modest protection of 44.0% (95% CI: 35.1; 51.6) against infection 14 to 90 days since vaccination with 2 doses mRNA-1273 [7]. The VE decreased quickly thereafter to 23.5% (95% CI: 16.4; 30.0) 91 to 180 days and 5.9% (95% CI: 0.04; 11.0) >270 days since vaccination [7]. Similar to our results, the VE against infection with the Omicron variant increased after the third dose [7]. They also observed similar VE estimates against hospitalization with the Omicron variant compared to our study with estimates of 84.5% (95% CI: 23.0; 96.9) and 99.2% (95% CI: 76.3; 100.0) after 2 or 3 doses, respectively [7].

The test-negative case–control study from England observed limited protection against symptomatic disease caused by the Omicron variant but reported that a third dose substantially increased protection [6]. Their observed 2-dose VE estimates against infection with the Omicron variant were 65.5% (95% CI: 63.9; 67.0) and 8.8% (95% CI: 7.0; 10.5) 2 to 4 weeks

and ≥25 weeks since vaccination, respectively, and were thus higher and waned faster after the third dose than observed in our study [6]. After the third dose, VE against infection with the Omicron variant was 67.2% (95% CI: 66.5; 67.8) and 45.7% (95% CI: 44.7; 46.7) 2 to 4 weeks and ≥10 weeks since vaccination, respectively [6]. An explanation as to the generally lower estimates in our study may be that we estimated VE against SARS-CoV-2 infection regardless of the symptom status.

## Strengths and limitations

The strengths of this study are the large scale of testing for SARS-CoV-2 including unlimited and easily accessible free-of-charge RT-PCR tests, as well as the ability to individually link data on all residents in Denmark across the nationwide high-quality registries. The high sensitivity (97.1%) and specificity (99.98%) observed for the RT-PCR test (21) minimize the risk of misclassification of the outcome. We cannot rule out that some SARS-CoV-2 infections might not be captured despite the mass testing strategy in Denmark. Some individuals might not have been tested or did not verify positive rapid antigen test or home-based test (only available during the Omicron wave) with a PCR test as recommended by the Danish Health Authorities. Furthermore, we were not able to discriminate between asymptomatic and symptomatic infections. In addition, a previous study has observed an inherent increased transmissibility of the Omicron sub-linage BA.2 [22], and we cannot rule out that the VE differs between the Omicron sub-linages. BA.1 and BA.2 were the most frequent Omicron sub-linages in Denmark during the Omicron study period (December 21, 2021 to January 31, 2022) [25]. BA.1 was most prevalent in the beginning of the included Omicron-dominated period. However, the prevalence of BA.2 has been increasing faster than BA.1 [25]. Due to the short BA.1-dominated period, it was not possible to separate the VE analysis by BA.1 and BA.2.

An effort was made to ensure equal access to COVID-19 vaccination for all Danish residents. This was done through an online booking system, special campaigns, offering vaccination in some workplaces, translating the information about COVID-19 vaccination to several languages, and arranging transport and pop-up vaccination for those who were not able to reach the vaccination clinics on their own. However, the populations initially prioritized for COVID-19 vaccination were the elderly, the most vulnerable citizens, and frontline healthcare workers, whereas the younger population was invited later. Therefore, it was not possible to estimate VE of 2 or 3 doses for both age groups in all defined periods. None had received the third dose in the Alpha-dominant period, and the majority of the oldest age group had already received their third vaccine dose in the Omicron-dominant period.

In general, nonrandomized studies assessing COVID-19 VE can easily be flawed [24], which may also apply to this study. Although we adjusted the Cox regression models for calendar time, age, sex, comorbidity, and geographical region, it is possible that other important confounders remained due to variations in test frequency and differences in behavior or adherence to COVID-19 guidelines between vaccinated and unvaccinated. Some vaccinated individuals may be more frequently tested (and thus more likely to document infection) if they are more health conscious compared with unvaccinated individuals. Conversely, some vaccinated individuals may be less frequently tested, if the vaccination reduces the severity of the infection and thus fewer infected people have symptoms. Furthermore, public health authorities may encourage more frequent testing for the unvaccinated while vaccinated individuals may become more heavily exposed to the virus after vaccination, if they feel liberated to engage in activities with more frequent and high-risk exposure [24]. This phenomenon of risk compensation decreases the medical benefit of vaccination [26]. However, some data have suggested little change in protective behavior early after vaccination [27]. During the course of the

study, the unvaccinated population became steadily smaller. Considering all of the above, it is possible that toward the end of the study, the remaining group of unvaccinated individuals, even after covariate adjustment, was substantially different from the rest of the population and noncomparable with respect to exposure to SARS-CoV-2 and progression to severe COVID-19. The Omicron analysis may thus have been affected more by such biases than the Alpha and Delta analyses given the diminishing size of the remaining unvaccinated population toward the end of 2021 and early 2022.

Overall, this study contributed with evidence of high vaccine protection against SARS-CoV-2 infection and importantly against hospitalization with the Alpha and Delta variants after 2 doses of BNT162b2 mRNA or mRNA-1273. These data support that a third dose provides good and relatively sustained protection against COVID-19 hospitalization with the Omicron variant and is necessary to maintain protection against infection.

## Supporting information

**S1 Table. STROBE Statement. Checklist of items that should be included in reports of cohort studies.**
(DOCX)

**S2 Table. Overview of results from previous studies.**
(DOCX)

**S3 Table. Unadjusted vaccine effectiveness of 2 doses BNT162b2 mRNA or mRNA-1273 against SARS-CoV-2 infection with the Alpha, Delta, or Omicron variant by age groups (12–59 years and 60 years or above).**
(DOCX)

**S4 Table. Unadjusted vaccine effectiveness of 3 doses BNT162b2 mRNA or mRNA-1273 against SARS-CoV-2 infection with the Delta or Omicron variant by age groups (18–59 years and 60 years or above).**
(DOCX)

**S5 Table. Unadjusted vaccine effectiveness of 2 doses BNT162b2 mRNA or mRNA-1273 against COVID-19 hospitalization following infection with the Alpha, Delta, or Omicron variant by age groups (12–59 years and 60 years or above).**
(DOCX)

**S6 Table. Unadjusted vaccine effectiveness of 3 doses BNT162b2 mRNA or mRNA-1273 against COVID-19 hospitalization following infection with the Delta or Omicron variant by age groups (18–59 years and $\geq$60 years).**
(DOCX)

## Acknowledgments

The authors are grateful to the Danish Health Data Authority for their help in defining the population. We would also like to thank the Department of Data Integration and Analysis at Statens Serum Institut for data management.

## Author Contributions

**Formal analysis:** Christian Holm Hansen.

**Methodology:** Christian Holm Hansen.

**Writing – original draft:** Mie Agermose Gram, Hanne-Dorthe Emborg, Nikolaj Ulrik Friis, Katrine Finderup Nielsen, Ida Rask Moustsen-Helms, Rebecca Legarth, Janni Uyen Hoa Lam, Manon Chaine, Aisha Zahoor Malik, Morten Rasmussen, Jannik Fonager, Raphael Niklaus Sieber, Marc Stegger, Steen Ethelberg, Palle Valentiner-Branth, Christian Holm Hansen.

**Writing – review & editing:** Astrid Blicher Schelde.

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
