## [Editor Report · Decision Letter 0]

13 Apr 2022

Dear Dr Gram, 

Thank you for submitting your manuscript entitled "Vaccine effectiveness against SARS-CoV-2 infection and COVID-19-related hospitalization with the Alpha, Delta and Omicron SARS-CoV-2 variants: a nationwide Danish cohort study" for consideration by PLOS Medicine.

Your manuscript has now been evaluated by the PLOS Medicine editorial staff and I am writing to let you know that we would like to send your submission out for external peer review.

Please re-submit your manuscript within two working days, i.e. by Apr 15 2022 11:59PM.

Kind regards,

Callam Davidson

Associate Editor

PLOS Medicine

---

## [Decision Letter · Decision Letter 1]

26 May 2022

Dear Dr. Gram,

Thank you very much for submitting your manuscript "Vaccine effectiveness against SARS-CoV-2 infection and COVID-19-related hospitalization with the Alpha, Delta and Omicron SARS-CoV-2 variants: a nationwide Danish cohort study" (PMEDICINE-D-22-01198R1) for consideration at PLOS Medicine. 

Your paper was evaluated by an associate editor and discussed among all the editors here. It was also discussed with an academic editor with relevant expertise, and sent to independent reviewers, including a statistical reviewer. The reviews are appended at the bottom of this email and any accompanying reviewer attachments can be seen via the link below:

[LINK]

In light of these reviews, I am afraid that we will not be able to accept the manuscript for publication in the journal in its current form, but we would like to consider a revised version that addresses the reviewers' and editors' comments. Obviously we cannot make any decision about publication until we have seen the revised manuscript and your response, and we plan to seek re-review by one or more of the reviewers. 

We hope to receive your revised manuscript by Jun 16 2022 11:59PM. Please email us (plosmedicine@plos.org) if you have any questions or concerns.

We look forward to receiving your revised manuscript. 

Sincerely,

Callam Davidson, 

PLOS Medicine

plosmedicine.org

In your Abstract, please combine the Methods and Findings sections into one section, “Methods and findings”.

In the last sentence of the Abstract Methods and Findings section, please describe the main limitation(s) of the study's methodology.

Please place citations in square brackets.

Many thanks for providing a STROBE checklist. Please update the checklist to use section names and paragraph numbers, rather than page/line numbers (as these are likely to change during the revision process).

Did your study have a prospective protocol or analysis plan? Please state this (either way) early in the Methods section.

The main text is missing a reference to Figure 3 (should the reference at line 187 read ‘3’ rather than ‘2’?).

Apologies if I missed it but I can’t see the corresponding asterisk in Table 1.

Figures 2/3 – Please update x-axis to read ‘oct’ rather than ‘okt’.

Tables 2-5: Please consider also including the unadjusted analyses as Supplementary Tables.

Please ensure journal abbreviations in the references list are consistent (e.g. 'The New England Journal of Medicine' in reference 2 ought to read ‘N Eng J Med’ as per reference 3).

Comments from the reviewers:

Reviewer #1: See attachment

Michael Dewey

Reviewer #2: The authors did an excellent job providing valuable vaccine effectiveness data on omicron as it compares to alpha and delta. 

1) we've noticed that hospitalization is a bit of a messy definition during omicron where many cases are incidentally infected with omicron or it aggravates an underlying medical problem. UKHSA saw a lower VE against hospitalization in younger cohorts specifically where you expect more incidental covid. see this summary https://linkinghub.elsevier.com/retrieve/pii/S0264410X22005230

2) can you please just include a sentence about how you dealt with persons with prior infection (were they included or excluded?) given the sample size i can't imagine it'll change the results dramatically especially for alpha/delta but just good to state.

3)if there is room, for your consideration is to add in the limitations that persons are assumed to be negative in this national cohort--undiagnosed persons or persons testing positive on home-based testing (if applicable in denmark) could bias the results. 

Excellent contribution to the literature. 

Minal Patel

Reviewer #3: Please find review comments in attached document. 

Reviewer #4: 

This study is of obvious relevance to public health, and the methods are clearly described. It is reassuring that the efficacy of two doses against hospitalised COVID-19 in those over 60 remains moderately high even with the Omicron variant. My comments focus mainly on the methods. 

1. For COVID-19 related hospitalisation as outcome, the case definition is based on first positive test from 14 days before to 2 days after admission. This excludes "probable health-care associated infections", defined by ECDC as cases having first positive test after more than 7 days in hospital. In the UK at the height of the epidemic "probable HAI" accounted for up to one-third of hospitalised COVID-19 cases and an even higher proportion of severe cases. If the proportion of "probable HCAI" was high in Denmark also, it would be important to report an analysis that includes this category in the case definition, even if only as supplementary material. 

2. In the Omicron wave, a high proportion of COVID-related hospitalisations may have been people hospitalised for other reasons,in whom a positive test was an incidental finding on admission. This may account for the apparently modest efficacy of 2 doses against Omicron-related hospitalisations in those aged under 60 years. The authors should discuss this possibility, and if possible try restricting the case definition to those whose hospitalisation was for COVID rather than with COVID if any further information such as main diagnosis or admitting specialty can be extracted from the electronic health record. 

3. For infection as outcome, the case definition is based on detection by on-demand testing. This may introduce selection bias: for instance the detection rate may vary with vaccination status for instance where regular testing and vaccination are mandatory in occupations such as health care. This is noted in the Discussion, though the statement that "we cannot exclude time-varying factors such as test frequency" is rather confusing as the problem is not that test frequency is time-varying. It would be useful to report testing rates by vaccination status over time. 

Most other studies of vaccine efficacy against infection have used test-negative control designs, though this may introduce other sources of bias. If the authors are able to link to negative test results as well as positive ones, it would be useful to report a test-negative control analysis for comparison with the cohort analysis. As the cohort design and the test-negative design have different biases, this would help to establish how robust are the findings with respect to infection as outcome. This applies only to the analyses with infection as outcome: for hospitalisation as outcome the case ascertainment is complete and detection bias does not arise.

[LINK]

---

## [Decision Letter · Decision Letter 2]

18 Jul 2022

Dear Dr. Gram,

Thank you very much for re-submitting your manuscript "Vaccine effectiveness against SARS-CoV-2 infection or COVID-19 hospitalization with the Alpha, Delta or Omicron SARS-CoV-2 variants: a nationwide Danish cohort study" (PMEDICINE-D-22-01198R2) for review by PLOS Medicine.

I have discussed the paper with my colleagues and the academic editor and it was also seen again by three reviewers. I am pleased to say that provided the remaining editorial and production issues are dealt with we are planning to accept the paper for publication in the journal.

[LINK]

We look forward to receiving the revised manuscript by Jul 25 2022 11:59PM.   

Sincerely,

Callam Davidson, 

Associate Editor 

PLOS Medicine

plosmedicine.org

Requests from Editors:

Line 47: Missing a full stop/space between last two sentences.

Line 54: ‘…better as compared to…’

Please consider splitting the bullet point at line 69 into two shorter bullet points. 

Please include key figures in your Author Summary where possible (for instance, sample size and main VE estimates (with 95% CI).

Lines 186 and 226: ‘were’ rather than ‘was’.

Please include covariates adjusted for in the legends of Figures 3 and 4.

Lines 454, 475, 502, and 549: Please remove the COI information from references 4, 8, 13 and 27.

For internet sources (e.g., references 11, 15, 23), please include date accessed.

Comments from Reviewers:

Reviewer #1: The authors have addressed all my points.

Michael Dewey

Reviewer #3: Thank you for addressing all of my comments. You've done a very nice job with this manuscript! 

Reviewer #4: I have no further comments on this paper.

[LINK]

---

## [Editor Report · Decision Letter 3]

26 Jul 2022

Dear Dr Gram, 

On behalf of my colleagues and the Academic Editor, Dr James Beeson, I am pleased to inform you that we have agreed to publish your manuscript "Vaccine effectiveness against SARS-CoV-2 infection or COVID-19 hospitalization with the Alpha, Delta or Omicron SARS-CoV-2 variant: a nationwide Danish cohort study" (PMEDICINE-D-22-01198R3) in PLOS Medicine.

PRESS

Sincerely, 

Callam Davidson 

Associate Editor 

PLOS Medicine